# Catamenial Pneumothorax as the First Expression of Thoracic Endometriosis Syndrome and Pelvic Endometriosis

**DOI:** 10.3390/jcm11051200

**Published:** 2022-02-23

**Authors:** Paola Ciriaco, Piergiorgio Muriana, Angelo Carretta, Jessica Ottolina, Massimo Candiani, Giampiero Negri

**Affiliations:** 1Department of Thoracic Surgery, Scientific Institute and University Vita-Salute San Raffaele, Hospital San Raffaele Milano, 20132 Milan, Italy; muriana.piergiorgio@hsr.it (P.M.); carretta.angelo@hsr.it (A.C.); negri.giampiero@hsr.it (G.N.); 2Department of Obstetrics and Gynecology, Scientific Institute and University Vita-Salute San Raffaele, Hospital San Raffaele Milano, 20132 Milan, Italy; ottolina.jessica@hsr.it (J.O.); candiani.massimo@hsr.it (M.C.)

**Keywords:** catamenial pneumothorax, thoracic endometriosis syndrome, pelvic endometriosis, videothoracoscopy, laparoscopy

## Abstract

Objective: The menstrual-related catamenial pneumothorax (CP) can be the first expression of thoracic endometriosis syndrome (TES), which is the presence of endometriotic lesions in the lungs and pleura, and pelvic endometriosis (PE). This study aims to analyze our experience with this specific correlation describing our multidisciplinary approach to CP. Methods: Hospital records of 32 women, operated for CP at our Department from January 2001 to December 2021 were reviewed. Surgical treatment consisted of videothoracoscopy and laparoscopy when indicated. Results: TES and PE were diagnosed in 13 (40.6%) and 12 (37.5%) women, respectively. The association of TES and PE was present in 11 cases (34%). Fifteen patients (46.9%) underwent laparoscopy, of which 11 concurrently with videothoracoscopy. Most of the patients affected had stage III–IV endometriosis (40.6%). All patients received hormonal therapy after surgery. Five patients with PE conceived spontaneously resulting in six live births. The mean follow-up was 117 ± 71 months (range 8–244). Pneumothorax recurrence occurred in six patients (18.8%). At present, all women are asymptomatic, with no sign of pneumothorax recurrence. Conclusions: CP might be the first expression of TES and/or PE. A multidisciplinary approach is advocated for optimal management of the disease.

## 1. Introduction

Catamenial pneumothorax (CP) is defined as recurrent spontaneous pneumothorax occurring between 24 h before and 72 h after the onset of menstruation. It has always been considered an unusual condition [1], but since its recognition has improved, its frequency is currently quoted as 23 to 30% of pneumothoraxes among women [2]. Endometriosis is a benign gynecologic disease affecting almost 10% of women of reproductive age. The presence of endometrial glands and stroma outside the uterus usually is found within the pelvic cavity, although rarely, the disease can extend to the extra pelvic region such as the lung and the pleura [3]. Thoracic endometriosis syndrome (TES) refers to the presence of endometriotic lesions in the lungs and pleura and comprises four clinical entities, namely catamenial pneumothorax, catamenial hemothorax, catamenial hemoptysis, or lung nodules [4]. CP is the most frequent expression of TES and pelvic endometriosis (PE) can be associated in 60% of cases [5]. Although CP in TES is often associated with PE, treatment is generally carried out in departments of thoracic surgery because of the clinical manifestations and few studies have considered a multidisciplinary approach with a gynecologic perspective [5,6,7].

In this series, we analyzed our experience of a multidisciplinary approach to catamenial pneumothorax as the first expression in TES, outlining the clinical and surgical features of the disease and the correlation with PE.

## 2. Materials and Methods

We retrospectively reviewed the hospital records of patients surgically treated for CP at the Department of Thoracic Surgery, from January 2001 to December 2021. Women of reproductive age between 18 and 45 years with pneumothorax were included in the study, excluding younger patients to avoid false-negative results [8]. Surgery was proposed to all patients who presented with (1) pneumothorax occurring between 24 h before and 72 h after the start of menses even at the first episode; (2) previous history of chest pain during menses; (3) recurrent pneumothorax; (4) history of pelvic endometriosis; (5) complicated pneumothorax.

Pneumothorax was considered an expression of TES based on macroscopic findings and/or histopathological reports that were either the presence of both endometrial stroma and glands or the presence of endometrial stroma alone with positive staining for CD10 and estrogens/progesterone receptors. PE was diagnosed based on the result of pelvic exploration by laparoscopy.

All patients who met the inclusion criteria underwent a preoperative gynecologic examination. Chest computed tomography scan or chest and/or abdomen magnetic resonance imaging was routinely introduced, as a preoperative workout, after 2010. Before this year, chest x-ray and pelvic ultrasound could be considered adequate for preoperative evaluation.

Videothoracoscopy was the surgical approach of choice and it was performed as previously described [9]. In brief, it was performed under general anesthesia with double-lumen bronchial intubation and three ports (5 or 12 mm). The lung was inspected for blebs or bulla. Signs of thoracic endometriosis were sought by inspection of the lung, pleura, and diaphragm. When diaphragmatic lesions were found, they were resected and in presence of diaphragmatic defects only, they were repaired through direct suture or plication of the muscle. Apical resection and apical pleurectomy were performed in patients who presented with blebs and/or bullae. Either mechanical or chemical pleurodesis was performed in all patients. Chemical basal pleurodesis with talc (1 g) was given to patients with diaphragmatic defects and lesions.

Laparoscopy was indicated by the gynecologist based on the patient’s history and/or instrumental findings. In patients with pain or infertility associated with PE, laparoscopic surgery was performed following the European Society of Human Reproduction and Embryology and American Congress of Obstetrics and Gynecology guidelines for managing women with endometriosis [3,10,11,12]. The possibility of a surgical approach was discussed with patients wishing to preserve fertility, patients poorly responding to medical treatment in terms of pain relief, and patients affected by endometriosis in the absence of previous ovarian surgery. The two procedures were performed either staged or simultaneously. In patients with PE, treatment was adapted to the clinical symptoms. Patients were then referred to our endometriosis outpatient clinic for a gynecologic follow-up. Owing to the absence of a histopathological report in the patients who did not undergo surgery, the presence of asymptomatic disease cannot be excluded in these patients. This group of patients was considered “potentially affected”. Different presentations of endometriosis were grouped according to the revised American Fertility Society classification system [12]. All the women signed written informed consent for surgery and to record their data for scientific purposes. The Ethics Committee of our Institution approved the study. Descriptive statistics were used to characterize the patient population. Continuous data with normal distribution are given as mean ± standard deviation.

## 3. Results

During the 20-year study, a total of 32 (17%) out of 183 women aged between 18 and 45 years, treated for pneumothorax at our Department of Thoracic Surgery, met the inclusion criteria. The mean age was 35.5 ± 6.3 years (range 21–46). Pneumothorax occurred within 24 h of the onset of menses in 19 patients, within 48 h in 7, and within 72 h in 6 patients. All patients presented with symptoms including chest pain, cough, shortness of breath, and hemothorax. Twenty-nine (90.6%) patients presented with more than one episode of pneumothorax during their life (mean number of episodes, 2.7 ± 3.5). The three surgically treated patients on the first episode of pneumothorax presented with hemothorax, a history of pelvic endometriosis, and one of them had prolonged air leaks for more than 48 h after chest tube placement. Four patients were already on hormonal treatment.

Pneumothoraxes were unilateral, recurrent, and right-sided in all patients; one (3%) patient had also a history of left-sided pneumothorax treated conservatively elsewhere. Chest computed tomography scan showed a diaphragmatic endometrial implant in three patients and magnetic resonance imaging was suggestive of diaphragmatic endometrial implant and herniation of the liver through a diaphragmatic defect in another patient.

At surgery, diaphragmatic defects were observed in 22 (68.7%) patients (Figure 1a) and 12 of them were found to have diaphragmatic implants histologically confirmed (Figure 1b).

One patient (3.1%) presented endometrial tissue in the resected bulla. Thirteen (40.6%) women were then diagnosed with TES.

Eleven patients (34.3%) also presented pleural (eight cases) and diaphragmatic (three cases) nodules suggestive for endometriosis that, instead, corresponded to hemorrhagic foci at pathology and were not counted as TES patients. Twenty-seven (84%) patients underwent chemical pleurodesis with talc. Surgical characteristics of patients are reported in Table 1.

Postoperative complications consisted of three cases of prolonged air leak (complications 9%). The mean duration of chest tube drainage was 4 ± 1 days (range 3–9).

After surgery, hormonal therapy was offered to all patients: three (9%) women were treated with progestins, the remaining 90.6% started estroprogestins.

The mean follow-up was 117 ± 71 months (range 8–244). Recurrence of pneumothorax occurred in six patients (18.8%) at two (2 cases), three (1 case), six, seven (2 cases), and nine months (1 case), and it was right-sided in all cases. Three patients were conservatively treated; two required chest tube positioning and one of these underwent chemical pleurodesis through the chest tube. One patient underwent a new surgical exploration with chemical pleurodesis.

At the time of recurrence, all patients, except one, were on hormonal treatment. After the recurrence, one patient, who had previously refused hormonal therapy and two patients on estrogen-progesterone therapy (continuous long term therapy) were put on gonadotropin-releasing hormone agonist for 6 months.

The follow-up was carried out based on periodic outpatient visits, both for thoracic surgery and gynecology. No patients were lost to follow-up.

At present, all women are asymptomatic with no sign of pneumothorax recurrence.

### 3.1. Presence of Pelvic Endometriosis

The diagnosis of PE preceded the diagnosis of TES in six patients (18.7%). Fifteen women (46.9%) underwent laparoscopy for the definitive histopathological diagnosis of PE that was confirmed in 12 patients (37.5%). The association of TES and PE was histologically confirmed in 11 patients (34%). In three cases of laparoscopy, PE was not confirmed. Videothoracoscopy and laparoscopy were staged 2 to 3 months apart in four cases and performed simultaneously in 11 cases. Pelvic endometriosis was in stage III–IV in 10 (86.6%) patients and stage I–II in two (16%).

Seventeen women (53%) had no indication for laparoscopy and did not exhibit clinical features suggesting the presence of pelvic endometriosis; nonetheless, endometriosis could not be excluded in these seven patients, owing to the absence of histological samples.

Laparoscopy procedures in PE are reported in Table 2.

### 3.2. Fertility Status

Seven patients with PE attempted to conceive. Five of them (71%) conceived spontaneously, resulting in six live births (three vaginal deliveries, three cesarean sections). Two women were unable to become pregnant and underwent infertility treatments without success. Patients were able to conceive after treatment for endometriosis.

## 4. Discussion

Pneumothorax defined as CP has always been considered a rare cause of spontaneous pneumothorax with a reported incidence in the past of 2.8–5.6% [13]. Recent series suggest that this incidence reached 41% of all pneumothoraxes in women probably due to an increased awareness of this pathology [1,14]. CP is identified as recurrent spontaneous pneumothorax occurring within 24 h before and 72 h after the onset of menses.

Endometriosis seems to play an important role in the development of CP considered one of the most frequent expressions of TES [15,16], and the association with PE has been reported in 20–70% of patients [7,16,17]. Although CP is now better recognized than in the past, diagnosis of endometriosis can be delayed [2]. In consideration of these data, the occurrence of pneumothorax in women of reproductive age should be managed as a possible expression of TES and/or PE [18].

Among our CP cases, we diagnosed 13 patients with TES and 12 with pelvic endometriosis. In four cases preoperative workouts were suggestive of endometriosis and six patients had already a diagnosis of pelvic endometriosis achieved by mean of laparoscopy of ultrasound. In the remaining patients, only the clinical history brought us to advocate a gynecological evaluation for possible PE. At the beginning of our experience, we did not routinely perform chest computed tomography scan or magnetic resonance imaging and this might be the reason why preoperative findings in the majority of patients were not indicative for TES and/or PE. The presence of a right pneumothorax in conjunction with menses, history, and/or symptoms related to endometriosis has always led us to propose a surgical option to the patients, in the strong suspicion of CP and eventually thoracic endometriosis.

Magnetic resonance imaging seems more accurate for the study of the abdomen, and it has been used in the preoperative evaluation for pelvic endometriosis in patients affected by TES [19]. In the recent literature, magnetic resonance imaging has emerged as the preferred radiological exam for the diagnosis of TES-related lesions. In particular, it showed high accuracy in the detection of diaphragmatic and pericardial endometriotic lesions [20] but it might also have a relevant role in the case of visceral, parietal pleural, or bronchopulmonary endometriosis [19,21]. Chest-abdomen magnetic resonance can provide all the information necessary to establish the correct path, whether it is surgical or not. Nevertheless, preoperative findings suggestive of endometriosis might not be certainly detected with each radiological method and have to be interpreted in association with clinical presentation [19].

One of the theories of CP is related to the hematogenous spread of endometrial tissue in the chest [22] but diagnosis of TES is not always verified by histopathological analysis, even though the patients present with a pathognomonic history [17]. Previous studies have reported lung parenchyma lesions in 13% to 30% of CP [2,23] in contrast with other reports that did not identify any pulmonary endometrial foci [7]. We have found only one incidental finding of an endometrial implant in a bulla that was not visible at the moment of surgery. This is probably due to the difficult localization of endometrial lesions in the collapsed lung during the one-lung ventilation and to the hypothetical different patterns of these lesions in the different times of menses [24,25].

Visceral pleural lesions were macroscopically observed in 68.8% of patients reported by Kumakiri [7]; however, none of those lesions was pathologically confirmed as endometriosis. We also observed brown, either visceral or diaphragmatic lesions in eleven patients that were not confirmed as endometrial lesions at pathology, though corresponding to hemorrhagic foci. Although this aspect may suggest endometriosis according to the literature, it cannot be definitively affirmed in such cases [2].

Diaphragmatic defects are frequently observed in TES pneumothoraxes suggesting that a trans-diaphragmatic passage of air coming from the uterus and tubes during pneumoperitoneum is probably the most frequent mechanism of pneumothorax in these patients [14]. Diaphragmatic defects in CP range from 29 to 66% of patients [22]. Some authors [24] report concomitant diaphragmatic defects and endometrial implants but the association is not widely observed [26]. In our series, 22 patients (68.7%) presented with diaphragmatic defects but only in 13 cases, endometrial implants were histologically confirmed. The disparity between the surgical pattern and histology can be the consequence of the fact that we do not perform routinely the resection of the diaphragm as other authors do [2,7,13] unless there are some visible implants, preferring the direct suture or plication, thus missing micro foci of endometrial implants at the pathological examination [26]. It has also been suggested that diaphragmatic defects might be the consequences of the evolution of full-thickness endometrial implants causing the formation of holes [23]. To avoid recurrences leaving behind small defects, we also performed localized chemical pleurodesis on the diaphragm.

Pleurodesis is recommended in CP surgical treatment [26]; we preferred talc pleurodesis in most of the cases because of the higher recurrence rate observed in mechanical pleurodesis [2,9,16].

Apical blebs and/or bullae might be the only findings in some cases of CP and are more frequently associated with the absence of TES [14]. This pattern can be the expression of the theory that the constrictor of bronchioles and vascular structure prostaglandin F2, which can be found in the plasma of some women during menstruation, may destroy alveolar tissue due to vasospasm, thus leading to pneumothorax [23]. In our series, we have found one incidental finding of endometrial focus in a bulla (solitary macroscopic finding), thus confirming the reported low incidence of TES in this group of patients [14]. We found that apical pleurectomy may be effective in cases of apical resections for the presence of blebs and/or bulla.

The strong association of CP not only with TES [18] but also with PE [3] emphasizes the need for optimizing a multidisciplinary approach to treatment with combined laparoscopy and thoracoscopy [7,24]. In our cohort, PE was diagnosed in 37.5% of CP and 84.6% of patients with TES, with 11 patients presenting with both TES and PE. This finding might suggest that TES can be considered a progression of endometriosis [18].

Laparoscopy makes possible the exploration of the pelvis, although the right lobe of the liver obscures the posterior portion of the diaphragm; thoracoscopy may overcome this problem allowing a complete exploration and treatment of the diaphragm as well as the lung, pleura, and the entire chest cavity [24].

Data about the use of combined or staged videothoracoscopy and laparoscopy are present in several trials [19] explaining how important it is to determine a standardized path for the treatment of TES. The choice of concomitant videothoracoscopy and laparoscopy surgeries seems also to lower delay in TES diagnosis [19].

The association of surgery and medical treatment is currently considered the standard of care for CP [5,7]. Gonadotropin-releasing hormone agonist is effective after surgery, in contrast to estrogen-progesterone therapy, which can lead to the recurrence of pneumothorax despite its positive effect in PE [4,5,25,26]. It is important to underline that therapy with gonadotropin-releasing hormone might be burdened by important side effects and must be suspended after 3–6 months, while estrogen-progesterone therapy can be continuous and suspended only if the patient is trying to get pregnant [3,10,25]. A more detailed analysis of the correlation between pneumothorax recurrence and hormonal treatment might be helpful to define the optimal therapy in our series [2,23,24,25].

Soriano et al. [5] have reported CP also as a possible predictor of infertility secondary to pelvic endometriosis, although in our experience, most of the patients who attempted conception conceived spontaneously (5 out of 7; 71%) after endometriosis treatment.

## 5. Conclusions

CP is still underdiagnosed despite the increased awareness in the medical community and improvement of detection [14]. CP is treated as a respiratory disease in the majority of cases, but it is now recognized that it can be the expression of TES and is frequently associated with PE [7,24,25,26,27]. Our data, which are consistent with the literature [7], suggest that all women with catamenial pneumothorax of reproductive age should be investigated for TES and/or PE. Thoracic surgeons and gynecologists should cooperate in the management of the disease although, in other hospitals organizations, patients with pneumothorax might also be evaluated at the beginning by pulmonologists. Surgical treatment includes videothoracoscopy and laparoscopy.

Limitations of the study are: (1) this is a retrospective study, so the conclusions are also the result of the experience gained over the years and (2) having included only CP patients in the study, the chance of having TES is higher compared to the female population with non-CP, which could be interesting to investigate.

In light of these considerations, we propose the following flowchart for a multidisciplinary approach in cooperation with gynecologists for optimal treatment of the disease (Figure 2).

## Figures and Tables

**Figure 1 jcm-11-01200-f001:**
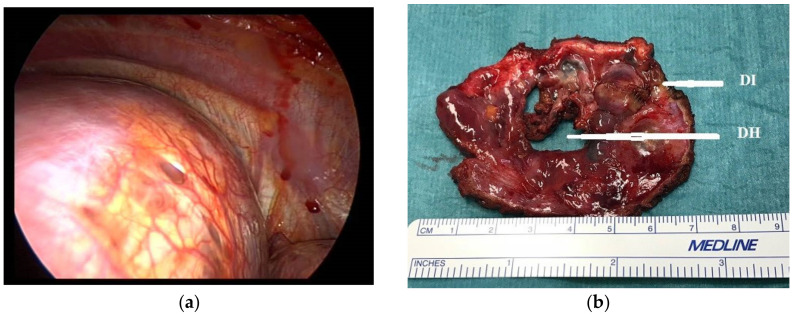
(**a**) Intraoperative finding of diaphragmatic defects; (**b**) surgical specimen showing a diaphragmatic hole (DH) and a diaphragmatic implant (DI) of endometriosis.

**Figure 2 jcm-11-01200-f002:**
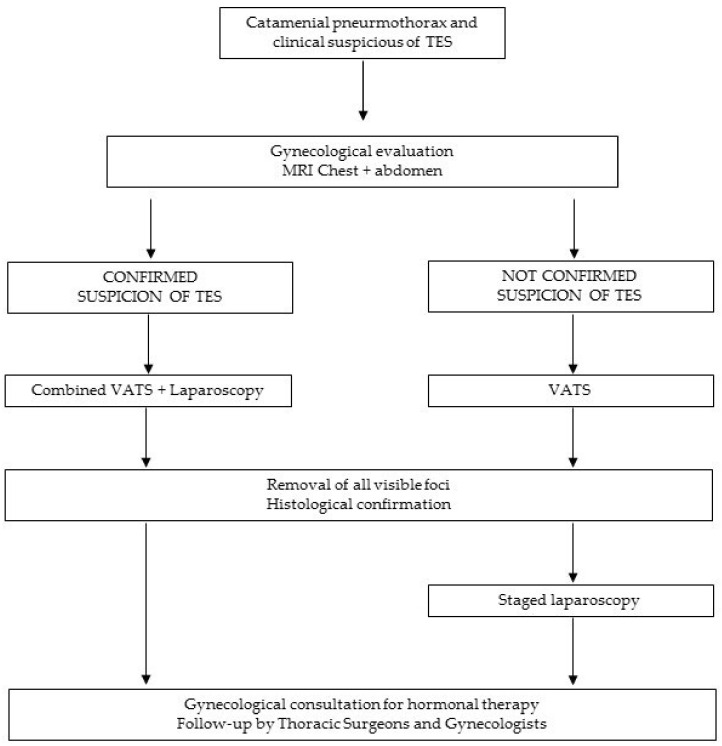
Flow chart for the multidisciplinary treatment.

**Table 1 jcm-11-01200-t001:** Surgical procedures performed in VATS.

Pts	Age (Years)	TES	PE	Surgical Approach	Intraoperative Thoracic Findings	Treatment
1	25	yes	yes	V + L	blebs + DL + DD	DR, DRR, AR, CP
2	26				blebs + DD	DR, AR, CP
3	42				DD + PL	DR, AP, CP
4	21				blebs	AR, AP, MP
5	28	yes	yes	V + L	blebs + DL + DD + PL	AR, AP, CP, PB, DR, DRR
6	41	yes	yes	V + L	blebs + DL + DD	DR, DRR, AR, CP
7	34				blebs + DD	DR, DRR, AR, CP
8	40				Bullae + PL	AP, CP, PB
9	31				blebs + DD	DR, DRR, AR, CP
10	40				bullae + PL	AR, CP, PB
11	34				Bullae + PL	AR, AP, PB
12	40				blebs + DD + PL	DR, AR, CP
13	42				blebs + DD	DR, AR, CP
14	26	yes	yes	V + L	DL + DD	DR, DRR, CP
15	46				blebs + DD + DL	DR, DRR, AR, CP
16	41				blebs + DD + DL	DR, DRR, AR, CP
17	29	yes	yes	V + L	bulla	AR, CP
18	32				blebs	AR, CP
19	39	yes	yes	V + L	blebs + DD + DL	DR, DRR, AR, CP
20	36	yes	yes	V + L	blebs + DD + DL	DR, DDR, AR, CP
21	44	yes	yes	V + L	DD + DL	DR, DDR, CP
22	40				blebs + DD + PL	DR, AR, CP
23	34		yes	V + L	blebs + DD	DR, AR, CP
24	36	yes	yes	V + L	DD + DL + blebs	AP, CP, DR, DRR
25	34	yes			DD + DL	DR, DRR
26	45				Blebs + DL	AR, DR, DRR, CP
27	29			V + L	Blebs + PL	AR, MP
28	36			V + L	blebs	AR, CP
29	40	yes	yes	V + L	blebs + DD + DL	AP, DR, DDR, CP
30	37			V + L	Blebs + PL	AP, MP, PB
31	37	yes			DD + DL	DR, DRR, CP
32	31	yes	yes	V + L	DL + DD	DR, DRR, CP

Legend: TES: thoracic endometriosis syndrome; PE: pelvic endometriosis; V: videothoracoscopy; L: laparoscopy; DL: diaphragmatic lesions; DD: diaphragmatic defects; PL: pleural lesion DR: diaphragmatic repair; DRR: diaphragmatic resection; AR: apical resection; CP: chemical pleurodesis; MP: mechanical pleurodesis; AP: apical pleurectomy; PB: pleural biopsy.

**Table 2 jcm-11-01200-t002:** Surgical procedures performed in laparoscopy in pelvic endometriosis.

Pts	Peritoneal Lesion Ablation	Lysis of Adhesions	Ovarian Cystectomy	Bowel Resection	Deep Infiltrating Endometriosis
1		Yes			yes
5		Yes			yes
6		yes			yes
14					yes
17	yes				yes *
19		yes	yes		yes
20		yes			yes
21			yes	yes	
23			yes		
24		yes	yes		
29	yes	yes			yes
32		yes	yes		

Legend: * umbilical endometriosis.

## Data Availability

The data presented in this study are available on request from the corresponding author. The data are not publicly available due to privacy reasons.

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
