# Peer review of "Catamenial Pneumothorax as the First Expression of Thoracic Endometriosis Syndrome and Pelvic Endometriosis"

_jcm, 2022, doi:10.3390/jcm11051200_

Round 1

Reviewer 1 Report

The authors report retrospective series of a rare condition. The report is interesting but few points need to be considered/clarified to make it even more relevant to the physicians who may encounter such a case:

  1. Since it is a surgical series, the true incidence of Catamenial pneumothorax cannot be judged. Hence it is relevant to mention in which female patients presenting with pneumothorax in emergency surgical options need to be considered besides timing of event in relation to onset of menstruation. Indications for surgical exploration may be stated.
  2. Since chest x-ray is available in all, even in earlier cases as mentioned, any clues for diagnosis besides the presence of pneumothorax should be mentioned very briefly.
  3. The authors report recurrence in about one-fifth of cases. All recurrences were within 9 months, though mean follow up is of >20 years. How was follow-up done. Were patients examined and evaluated physically for this study, or was it a telephonic follow up or based on hospital records. It is likely that patients presenting with the condition over 20 years are being reported (as mentioned) and follow up regarding recurrence is being reported only on those who presented again to this particular hospital. If so, it should be stated clearly that follow up was based on presentation at this hospital only as patients may have shifted to another area and got treatment at another facility.
  4. Please mention what was done for management of recurrences as most were already on hormonal therapy. Did any patient require repeat surgery/pleurodesis? What further medical therapy was given: with dose and duration.
  5. Was consent obtained from all patients as reported? If so how: telephonically or written consent. Or else ethical approval was obtained to study and report based on patient records.

English language correction may be appropriate at few places, eg.

Line 34, page 1: “….since his recognition………….”, replace ‘his’ with ‘its’  

Line 153, page 5: “Pregnancies were obtained after the …..” Can be changed to “Patients were able to conceive….  or become pregnant….”

Author Response

Response to Reviewer 1 Comments

Point 1:  Since it is a surgical series, the true incidence of Catamenial pneumothorax cannot be judged. Hence it is relevant to mention in which female patients presenting with pneumothorax in emergency surgical options need to be considered besides timing of event in relation to onset of menstruation. Indications for surgical exploration may be stated.

Response 1: To comply with Point 1 the following statements were added:

Materials and Methods page 2 , lines 55-56 “…with pneumothorax, were included in the study”

Materials and Methods page 2, lines 56-60 “Surgery was proposed to all patients who presented with 1) pneumothorax occurring between 24h before and 72h after the start of menses even at the first episode; 2) a previous history of chest pain during menses; 3) recurrent pneumothorax; 4) history of pelvic endometriosis; 5) complicated pneumothorax”

Point 2:  Since chest x-ray is available in all, even in earlier cases as mentioned, any clues for diagnosis besides the presence of pneumothorax should be mentioned very briefly.

Response 2: To comply with Point 2 the following statement was added:

Discussion, page 6, lines 216-219 “The presence of a right pneumothorax in conjunction with menses, history, and/or symptoms related to endometriosis has always led us to propose a surgical option to the patients, in the strong suspicion of CP and eventually thoracic endometriosis”

Point 3:  The authors report recurrence in about one-fifth of cases. All recurrences were within 9 months, though mean follow up is of >20 years. How was follow-up done. Were patients examined and evaluated physically for this study, or was it a telephonic follow up or based on hospital records. It is likely that patients presenting with the condition over 20 years are being reported (as mentioned) and follow up regarding recurrence is being reported only on those who presented again to this particular hospital. If so, it should be stated clearly that follow up was based on presentation at this hospital only as patients may have shifted to another area and got treatment at another facility.

Response 3: To comply with Point 3 the following statement was added:

Results page 5, lines 162-163 “The follow-up was carried out based on periodic outpatients’ visits, both for thoracic surgery and gynecology. No patients were lost to follow-up”.

Point 4:  Please mention what was done for management of recurrences as most were already on hormonal therapy. Did any patient require repeat surgery/pleurodesis? What further medical therapy was given: with dose and duration.

Response 4: To comply with Point 4 the following statements were added:

Results page 5, lnes 154-157. “Three patients were conservatively treated, two required chest tube positioning and one of these underwent chemical pleurodesis through the chest tube. One patient underwent a new surgical exploration with chemical pleurodesis.

Results, page 5, lines 158-161. “After the recurrence, one patient, who had previously refused hormonal therapy, and two patients on estrogen-progesterone therapy (continuous long term therapy) were put on gonadotropin-releasing hormone agonist for 6 months”

Point 5:  Was consent obtained from all patients as reported? If so how: telephonically or written consent. Or else ethical approval was obtained to study and report based on patient records.

Response 5: To comply with Point 5 the following statement was added:

Material and Methods page 3, lines 95-97 “All the women signed written informed consent for surgery and to record their data for scientific purposes. The Ethics Committee of our Institution approved the study”

English language correction

English language correction may be appropriate at few places, eg.

Line 34, page 1: “….since his recognition………….”, replace ‘his’ with ‘its’  

Line 36, page 1 “….since his recognition.”,  has been replaced with ‘its’  

Line 153, page 5: “Pregnancies were obtained after the …..” Can be changed to “Patients were able to conceive….  or become pregnant….”

Line 191, page 6 “Pregnancies were obtained after the …..” has been changed to “Patients were able to conceive…. 

Reviewer 2 Report

I find this article interesting and thought provoking. Whilst it is a retrospective analysis, the authors should state ethical approvals and whether informed consent was waived, and if consent was obtained for the pictures

The article is well written - the grammar is good, and results well described. The literature is assessed appropriately. 

However, if possible, I would like the authors to re-assess their findings- surely not all patients came directly to surgery, and often in accepted practice, patients will have a thorough history and exam, and have hormonal treatment before surgery.....It might be an idea that they describe how the patients presented, how they were initially treated, when was gynae opinions sought and when was surgery decided upon? If they had surgery at their first pneumothorax, then that needs to be justified, as most first pneumothoraces are usually managed with tube drainage and then followed up for recrurrence. this would improve the flow of the article 

Can i also suggest that they insert a flow chart to suggest what the current course of action in a young woman presenting with pneumothorax should be- not all need surgery, and most will be seen in places not affiliated with cardiothoracics- it is still all about the history and the examination- and that flow chart should also depict who what their local practice is. 

Could they also comment on the length of hormonal treatment? as in my experience, patients get significant side effects from those and treatment often needs to be stopped. 

I would be happy to review again. 

Author Response

Response to Reviewer 2 Comments

Point 1:  However, if possible, I would like the authors to re-assess their findings- surely not all patients came directly to surgery, and often in accepted practice, patients will have a thorough history and exam, and have hormonal treatment before surgery.....It might be an idea that they describe how the patients presented, how they were initially treated, when was gynae opinions sought and when was surgery decided upon?

Response 1: To comply with Point 1 the following sentences were added:

Results page 3, lines 107-110 “The three surgically-treated patients on the first episode of pneumothorax presented with hemothorax, a history of pelvic endometriosis and one of them had prolonged air leaks for more than 48h after chest tube placement. Four patients were already on hormonal treatment”

Materials and Methods page 2, lines 66-67 “All patients who met the inclusion criteria underwent a preoperative gynecologic examination”

Point 2:  If they had surgery at their first pneumothorax, then that needs to be justified, as most first pneumothoraces are usually managed with tube drainage and then followed up for recrurrence. this would improve the flow of the article.

Response 2: See response of Point 1

Point 3:  Can I also suggest that they insert a flow chart to suggest what the current course of action in a young woman presenting with pneumothorax should be- not all need surgery, and most will be seen in places not affiliated with cardiothoracics- it is still all about the history and the examination- and that flow chart should also depict who what their local practice is. 

Response 3: To comply with Point 3 a flow chart of the multidisciplinary treatment has been added at the end of the Discussion section Page 8

Point 4:  Could they also comment on the length of hormonal treatment? as in my experience, patients get significant side effects from those and treatment often needs to be stopped. 

Response 4: To comply with Point 4 the following sentences were added

Discussion page 7, lines 285-288 “It is important to underline that therapy with gonadotropin-releasing hormone might be burdened by important side effects and must be suspended after 3-6 months, while estrogen- progesterone therapy can be continuous and suspended only if the patient is trying to get pregnant”

Round 2

Reviewer 2 Report

Thank you for this revised version: I thank the authors for their efforts, but again this does not sit right with me. In most instances, pneumothoraces will be managed by respiratory physicians at first presentation and only referred to surgeons when a prolonged air leaks happens. So while the inclusion criteria makes some sense, I dont think it is applicable beyond this study and thus this has limited value. In most centres, I think the approach would be immediate treatment by physicians, elucidation of diagnosis, then referral to a gynaecologist with a special interest in endometriosis and then surgery. The addition of the MRI component also needs to be described, as the MRI findings could be totally diagnostic and obviate need for surgery. 

I think the revision has made the article more confusing. I think what you are describing is all the patients you operated on who had thoracic endometriosis, rather than those who you selected. The authors will also then need a numerator and a denominator. 

I am sorry but I would like a new approach to this article ( the findings of which are publishable, but in a different format)- thanks
